



# Measurement Report: Spatiotemporal variability of peroxy acyl nitrates (PANs) over Mexico City from TES and CrIS satellite measurements

Madison J. Shogrin[1], Vivienne H. Payne[2], Susan S. Kulawik[3], Kazuyuki Miyazaki[2], and Emily V. Fischer[1]

[1]Colorado State University, Department of Atmospheric Science, Fort Collins, CO, USA
[2]Jet Propulsion Laboratory, California Institute of Technology, Pasadena CA, USA
[3]Bay Area Environmental Research Institute, Petaluma, CA, USA

*Correspondence to*: Madison J. Shogrin (madison.shogrin@colostate.edu)

**Abstract.** Peroxy acyl nitrates (PANs) are photochemical pollutants with implications for health and atmospheric oxidation capacity. PANs are formed via the oxidation of non-methane volatile organic compounds (NMVOCs) in the presence of nitrogen oxide radicals ($NO_x = NO + NO_2$). While urban environments are large sources of PANs, in-situ observations in urban areas are limited. Here we use satellite measurements of PANs from the Tropospheric Emission Spectrometer (TES) and the Suomi National Polar-orbiting Partnership (S-NPP) Cross-Track Infrared Sounder (CrIS) to evaluate the spatiotemporal variability of PANs over and around Mexico City. Monthly mean maxima in PANs over the Mexico City Metropolitan Area (MCMA) occur during spring months. This time of year coincides with a peak in local photochemistry and more frequent air stagnation. Local fire activity also typically peaks between February and May, which leads to strong interannual variability of PANs over the MCMA. We use S-NPP CrIS data to probe the spatial outflow pattern of PANs produced within urban Mexico City during the month with the largest mixing ratios of PANs (April). Peak outflow in April occurs to the northeast of the city and over the mountains south of the city. Outflow to the NW appears infrequent. Using observations during 2018 versus 2019, we also show that PANs were not significantly reduced during a year with a significant decrease in $NO_x$ over Mexico City. Our analysis demonstrates that the space-based observations provided by CrIS and TES can increase understanding of the spatiotemporal variability of PANs over and surrounding Mexico City.

## 1 Introduction

Megacities are large metropolitan areas with greater than ten million residents (Gurjar & Lelieveld, 2005). As of 2018, roughly 55 percent of the world's population resided in urban areas and this is expected to increase to over 60 percent by 2030 (UN/DESA, 2018). Megacities have become important sources of air pollutants, and these emissions contribute to regional and global trace gas budgets as plumes emitted in megacities are redistributed away from source regions, with implications for regional air quality and photochemistry (Gurjar & Lelieveld, 2005; Lawrence et al., 2007; Madronich, 2006; Mage et al., 1996; Molina et al., 2007).

Mexico City is one of the world's largest megacities with a population greater than 20 million and is well known for its high levels of pollution and reduced visibility (Molina and Molina, 2002; Molina et al., 2009). The Mexico City Metropolitan Area (MCMA) is situated at a high altitude in the tropics (2240 m; 19.43° N, 99.13° W), and it is surrounded on three sides by mountains with a wide-open basin to the north and a mountain passage to the southeast (Chalco passage) (Fig. 1). The MCMA is particularly prone to poor air quality given the regional topography, frequently weak synoptic forcing, and emissions from a variety of sources (see Molina et al., 2007 and references within).





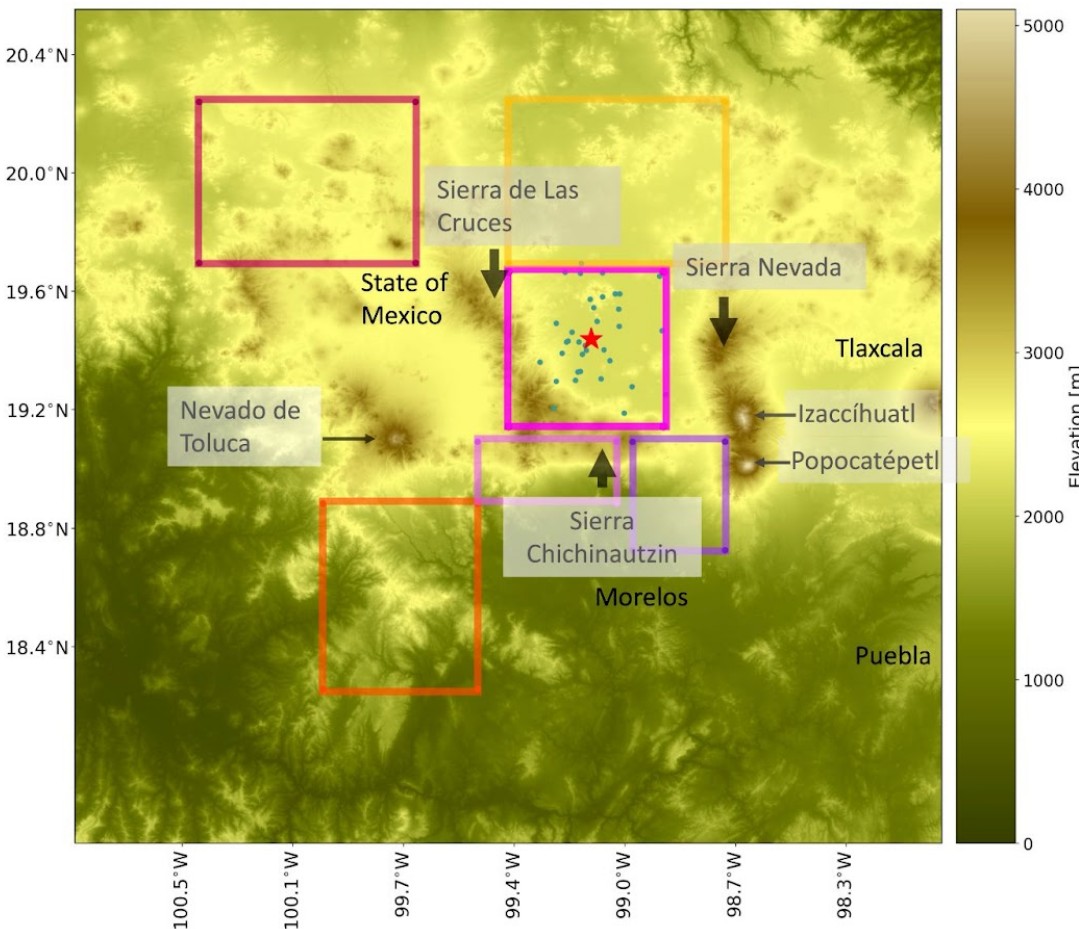

**Figure 1.** Elevation map of the Mexico City region. The red star denotes the center of Mexico City and the pink box represents the area averaged to create monthly means used throughout the analysis. Colored boxes represent regions of potential pollutant outflow further shown in Fig. 4.

During the mid 1980s and 1990s, Mexico City was ranked as the most polluted megacity in the world with all criteria pollutants exceeding air quality standards for human health (UNEP & WHO, 1992). Pollutants of concern include ozone ($O_3$), fine particulate matter (PM), nitrogen oxides ($NO_x = NO + NO_2$), and volatile organic compounds (VOCs). In addition to monitoring by the Mexico City atmospheric monitoring system (Sistema de Monitoreo Atmosférico or SIMAT), there have been a number of other efforts to observe and attribute regional photochemistry using in situ measurements. Analysis of field campaign datasets (e.g, MCMA-2003 and MILAGRO-2006; Molina et al., 2007; 2010) indicate that $O_3$ formation is often VOC-limited in the urban core and largely $NO_x$-limited in the surrounding area (Lei et al., 2007, 2008; Tie et al., 2007). The extent of $NO_x$-versus-VOC limited regions is dependent on meteorological conditions (Lei et al., 2008; Song et al., 2010).

Peroxy acyl nitrates (PANs) are an important photochemically produced species that may play an important role in diagnosing aspects of photochemistry within the MCMA and determining the scale over which the MCMA impacts atmospheric composition. PANs are formed alongside $O_3$ in polluted environments when non-methane volatile organic compounds (NMVOCs) are oxidized in the presence of $NO_x$ (Singh and Hanst, 1981; Fischer et al., 2014; Gaffney et al.,



1989; Roberts, 2007; Singh et al., 1986). The high elevation (Fig. 1) and tropical latitude of the MCMA makes for intense sunlight to the area year round, supporting efficient production of PANs given an abundance of precursors (Bravo et al.,
1989; MARI, 1994; Emmons et al., 2010; Fast & Zhong, 1998; Lei et al., 2007, 2008; citations from Marley et al., 2007; Streit & Guzman, 1996; Tie et al., 2007). Substantially elevated PANs have been observed within Mexico City (Bravo et al., 1989; MARI, 1994; Emmons et al., 2010; Fast & Zhong, 1998; Gaffney et al., 1999; Lei et al., 2007, 2008; Marley et al., 2007; Streit & Guzman, 1996; Tie et al., 2007). example, maximum daily mixing ratios in 1997 reached ~34 ppbv (Gaffney et al., 1999) and a maximum of ~8 ppbv was observed in 2003 (Molina et al., 2007; 2010). PANs cause a suite of health
effects (Altshuller, 1978; Shepson et al., 1987; Smith, 1965; Taylor, 1969; Kleindienst et al., 1990; Vyskocil et al., 1998) and they are also critical to understanding $O_3$ production. For example, $O_3$ production can continue in outflow regions of the MCMA as PANs act as a reservoir and source of $NO_x$ (Mena-Carrasco et al., 2009). Production of PANs can also help understand the photochemistry of this area (Sillman & West, 2009). PAN concentrations can be used to gauge the effectiveness of $O_3$-control strategies in urban areas, since PANs act as an indicator of regional photochemical activity
(Gaffney et al., 1989). Comparison of field campaign observations from 2003 and 2006 to observations from 2014 and 2019 indicate that there have been major changes in $O_3$-relevant VOC sources and oxidized nitrogen chemistry since the earlier studies (e.g., Lei et al., 2007, 2008; Tie et al., 2007; Zavala et al., 2020) This points to a need for continued and expanded observations of photochemically-relevant species in this region, including PANs.

Here we present new satellite observations of PANs over Mexico City. We leverage measurements from the Aura Tropospheric Emission Spectrometer (TES) and the Suomi-National Polar-orbiting Partnership (S-NPP) Cross-Track Infrared Sounder (CrIS), as well as other complementary satellite and ground-based observations. The satellite data demonstrates the regional seasonality and export pathways for this important global pollutant.


## 2 Methods

### 2.1 Satellite observations of PANs

PANs have absorption features in the thermal infrared that can be readily measured with spaceborne spectrometers. Limb-sounding satellite observations have provided global-scale information on PAN in the upper troposphere and lower
stratosphere with high vertical resolution and sensitivity (Glatthor et al., 2007; Moore & Remedios, 2010; Pope et al., 2016; Tereszchuk et al., 2013; Ungermann et al., 2016; Wiegele et al., 2012), but the limb-viewing geometry is not well suited to evaluation of urban influences on the free-troposphere. Nadir-viewing observations of PAN have been reported from TES (Payne et al., 2014), the Infrared Atmospheric Sounding Interferometer (IASI) (Franco et al., 2018) and CrIS (Payne et al., 2022). These nadir observations have shown large enhancements in PAN associated with fires (Alvarado et al., 2011;
Clarisse et al., 2011; Juncosa Calahorrano et al., 2021) and have so far been used to shed new light on the role of fires, PAN precursor emissions and dynamics on the global distribution of PAN and on long-range transport of $O_3$ (Fischer et al., 2018; Jiang et al., 2016; Payne et al., 2017; Zhu et al., 2015, 2017). The TES record includes a set of targeted special observations over megacities between 2013 and 2015 (Cady-Pereira et al., 2017). Further details are provided below. The spatial and temporal coverage routinely provided by the IASI and CrIS meteorological sounders offers rich opportunities for
examination of wider spatial context and long-term variation of PAN around megacities.

For various reasons, direct comparisons of the TES, IASI and CrIS products discussed above would be challenging. The three instruments have different noise characteristics, spectral coverage and spectral resolution, which have affected choices made for different algorithms. The TES PAN retrievals utilized the PAN spectral feature centered around 1150 cm$^{-1}$, while
the IASI observations used a larger spectral range that includes multiple PAN spectral features. The CrIS observations have made use of a spectral feature centered around 790 cm$^{-1}$. This spectral feature appears in the IR spectra of all PANs at essentially the same frequency. Thus, the CrIS measurements reported are for all PANs (i.e., they include propionyl peroxy nitrate (PPN; $CH_3CH_2C(O)OONO_2$), methacryloyl peroxy nitrate (MPAN; $CH_2C(CH_3)C(O)OONO_2$), etc.) in addition to peroxy acetyl nitrate (PAN; $CH_3C(O)O_2NO_2$). The IASI PAN product described by Franco et al. (2018) utilizes a neural
network approach to retrieve PAN total column densities, while the TES and CrIS products cited above utilize an optimal



estimation approach to retrieve profiles of volume mixing ratio (VMR), which have then been used to calculated free-tropospheric averages. The TES and CrIS PAN products share a common algorithm heritage, though there are some important differences, as discussed above and in the sections below. In this study, we use the targeted TES megacity observations to provide a view of PAN over Mexico City in the context of other selected megacities worldwide, and use
CrIS observations to focus in on the Mexico City Metropolitan Area and immediate surroundings.

## 2.2 TES observations

TES is a nadir-viewing Fourier transform spectrometer capable of measuring thermal infrared radiances at high spectral resolution (0.06 cm$^{-1}$). TES is one of four instruments aboard the NASA Aura satellite, which flies in a sun-synchronous
polar orbit in the NASA Afternoon train (A-train) constellation with equatorial overpass times of 01:30 and 13:30 LT. Aura was launched in July 2004, and the TES instrument took science measurements between September 2004 and January 2018.

A full description of the TES PAN retrieval algorithm is provided in Payne et al. (2014) and key details are summarized in Fischer et al. (2018). We use the publicly available TES v7 Level 2 product (NASA/LARC/SD/ASDC, 2017). On a single
footprint basis, TES is sensitive to elevated PAN (detection limit ~0.2 ppbv) in the free troposphere with uncertainties between 30-50 %. The peak sensitivity for TES PAN is typically between 400 and 800 hPa (Payne et al., 2014). In general, TES PAN retrievals have about one degree of freedom, meaning that the retrievals do not contain information on the vertical distribution of PAN. The TES PAN retrievals are performed using an optimal estimation approach, with the state vector expressed in log VMR. One impact of this is that the degrees of freedom for signal (DOFS) depends on the amount of PAN
present. For the analysis presented below, we use a tropospheric average from 825 hPa to 215 hPa for retrievals with DOF > 0.6. This criterion ensures that we only include retrievals dominated by signal in the measurement rather than signal from the prior value. We also only use retrievals where the PANs desert quality flag > 0.95, as suggested by Payne et al. (2014) to avoid issues with a silicate feature that occurs in the surface emissivity for rocky/sandy surfaces and happens to coincide with the location of the 1150 cm$^{-1}$ PAN spectral feature.

Here we use TES "transect" special observations over megacities collected between January 2013 and December 2015. TES has three methods of observation: global survey, step-and-stare mode, and transect mode. Global survey mode is the nominal observation strategy for TES in which the instrument makes periodic observations along the satellite track spaced ~200 km apart. In step-and-stare mode the instrument takes nadir measurements every 40 km along the satellite track for a specified
latitude range, and in transect mode TES takes 20 consecutive scans spaced ~12 km apart. An observation strategy focusing on 19 of the world's megacities was introduced in January 2013 and operated through December 2015. This sampling strategy reduced the number of sample points globally, but increased the number of retrievals over the selected 19 megacities. Each TES megacity transect consists of 20 footprints, each footprint 5 by 8 km in size, spaced 12 km apart resulting in relatively dense coverage along the orbit track over a limited area, providing a chemical snapshot of each
megacity roughly every 2 weeks. For each of these megacities, we calculated the mean tropospheric PAN over all TES transect observations that pass the quality screening criteria over the 2013 to 2015 observation period. This provides an overall picture of "high-PAN" vs "low-PAN" cities for this set of observations. Figure 2 ranks megacities by the mean detected tropospheric PAN during the TES megacity sampling period. During this period, the highest mean PAN was in Mexico City (average tropospheric PAN mixing ratio of 0.35 ppbv). Thus, the MCMA serves as our first case with new CrIS
observations (described next) given the abundance of PAN, a long history of poor air quality, and the availability of complementary datasets.
.





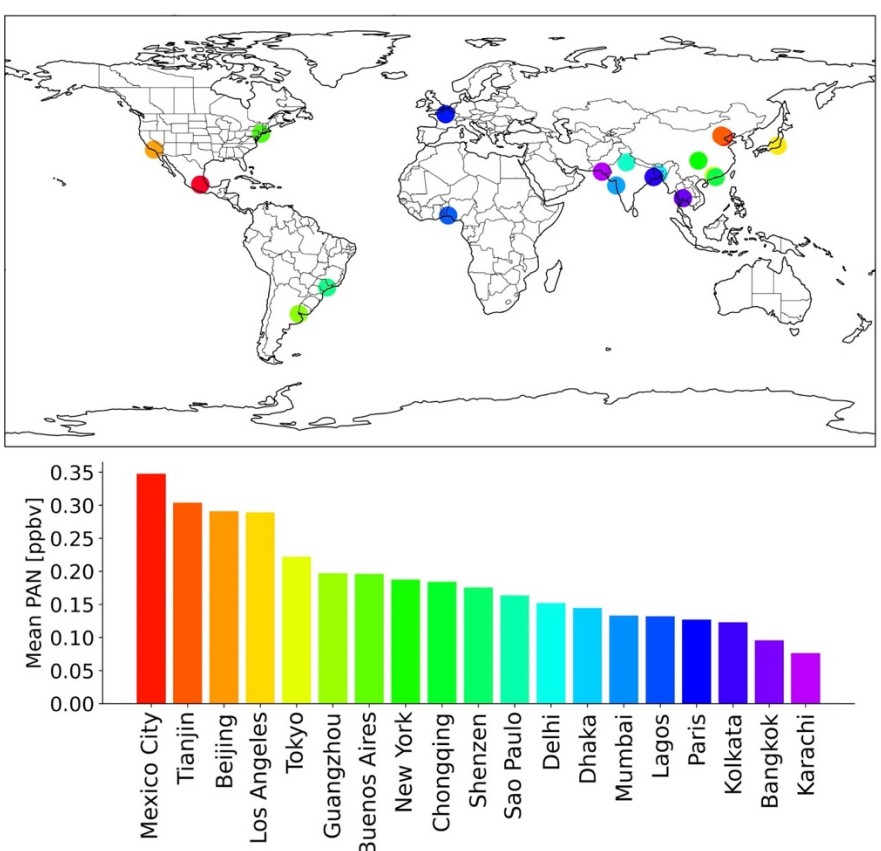


**Figure 2.** Ranked mean PAN in TES megacity transects collected between 2013 and 2015.

## 2.3 CrIS observations

CrIS is a nadir viewing Fourier transform spectroradiometer measuring thermal infrared radiances with high spectral resolution (0.625 cm$^{-1}$). CrIS instruments are currently flying on the Suomi National Polar-Orbiting Partnership (S-NPP) satellite and on the National Oceanic and Atmospheric Administration (NOAA-20/JPSS-1) satellite as part of the Joint Polar Satellite System (JPSS). Here we use CrIS on S-NPP. CrIS is one of five instruments aboard S-NPP which flies in a sun-synchronous polar orbit with equatorial overpass times at 01:30 and 13:30 LT. CrIS provides measurements at 30 crosstrack positions, each with a 3 x 3 array of fields of view (FOVs), where each field of view has a diameter of 15 km at nadir.

Processed CrIS data provides calibrated and geolocated Level 1B spectra in three bands: 60-1095 cm$^{-1}$ (longwave), 1210-1750 cm$^{-1}$ (mid-wave), and 2155-2550 cm$^{-1}$ (shortwave).

The PANs feature used in this analysis is located at 790 cm$^{-1}$. CrIS retrievals are processed using the MUlti-SpEctra, MUlti-SpEcies, MUlti-Sensors (MUSES) retrieval software (Fu et al., 2013, 2016, 2018; Worden et al., 2019) which builds on the

optimal estimation algorithm developed for Aura-TES (Bowman et al., 2006, Beer et al., 2001). One difference from the TES PAN retrieval algorithm is that the retrievals are done in linear VMR (as opposed to log). Further details of the CrIS PANs retrieval can be found in Payne et al. (2022) This analysis used the L2 product variable XPAN800, which is the column volume mixing ratio between 825 and 215 hPa. CrIS sensitivity to PANs peaks in the free troposphere (~680 hPa)





and decreases rapidly near the surface. Payne et al. (2022) show validation of CrIS PANs observations against aircraft observations over the remote oceans. The validation suggests a single sounding uncertainty of around 0.08 ppbv that reduces with averaging to an approximate floor of 0.05 ppbv and demonstrates the ability of CrIS PANs retrievals to capture variation in the "background" PANs over remote regions.

CrIS-MUSES single-FOV retrievals of PANs, as well as temperature, water vapor (H$_2$O), deuterated water vapor (HDO), O$_3$, carbon monoxide (CO), methane (CH$_4$) and ammonia (NH$_3$) are being processed routinely under the NASA Tropospheric Ozone and Precursors from Earth System Sounding (TROPESS) project and are publicly available via the Goddard Earth Sciences Data and Information Services Center (GES DISC). The TROPESS datasets at the GES DISC include the forward stream, which incorporates both S-NPP and JPSS-1 CrIS data (Bowman, 2021a; Bowman et al., 2021b). There are also plans to release a "reprocessing" dataset, for which the long-term record will be processed with a uniform algorithm version. Due to constraints associated with algorithm speed and the volume of data from the CrIS meteorological sounder(s), the TROPESS project is not currently processing all available CrIS radiances. The TROPESS CrIS forward stream data are subsampled using a grid sampling approach where the region is divided into 0.8 ° latitude x 0.8 ° longitude grid boxes and the single, center-most target within each box is selected to be included in the dataset. The forward stream dataset provides both day and night time coverage. The TROPESS datasets also include so-called "special collections" where the sampling may be tailored to address a particular scientific study (or studies). Data for this work were processed as part of a special collection, using the v1.12 of the MUSES algorithm, to provide retrievals for all S-NPP CrIS field of views (FOVs) for the date range January 2016 to May 2021 in 2 by 2 ° latitude/longitude boxes centered on specific megacities. The list of cities was chosen to match those included in the TES megacity transect set. Note that since the S-NPP and JPSS-1 CrIS instruments are essentially identical, targeted processing of JPSS-1 CrIS over megacities is also possible, but has not yet been performed at the time of submission of this work.

CrIS CO data from the same special collection was also incorporated into this analysis as a supplementary dataset to contextualize CrIS PANs observations. For this analysis, we use a tropospheric average of CrIS CO data between 825 and 215 hPa. CrIS can be more sensitive to CO than to PANs in the lower troposphere (Juncosa Calahorrano et al., 2021). Details of the CrIS CO retrieval can be found in Fu et al. (2016) and CrIS CO validation against aircraft observations in Worden et al. (2022). Note that the TROPESS CrIS CO forward stream is also available at the GES DISC (Bowman et al., 2021c; Bowman et al., 2021d).

This study used S-NPP CrIS data from January 2016 to May 2021. To be included in this analysis, CrIS retrievals had to meet the following criteria:

1. Radiance Residual RMS (Root Mean Square of the standard deviation) < 5
2. Radiance Residual Mean < 2
3. Residual Norm Final < 5
4. Quality flag in PANs files = 1
5. Quality flag in water vapor files = 1
6. B - A > -0.15 (see Eq. 1 and 2 below)

CrIS data was filtered based on the radiance files using the following equations to check if there is an improvement in the standard deviation, where NESR is the Noise Equivalent Spectral Radiance:

$$A = \frac{\text{Standard Deviation(Radiance Fit} - \text{Radiance Observed)}}{\text{NESR}}, \qquad (1)$$

$$B = \frac{\text{Standard Deviation(Radiance Fit Initial} - \text{Radiance Observed)}}{\text{NESR}}, \qquad (2)$$



### 2.4 Additional datasets

Given the complexity of PANs chemistry and the variety of sources that contribute to PANs abundances, we incorporate several other datasets into our analysis to contextualize the PANs observations.


Surface $O_3$ and $NO_x$ data included here are from the environmental government network of Mexico City (Red Automática de Monitoreo Atmosférico; RAMA) (RAMA, 2022). We include monthly means of day time averages for all of the surface sites within the Mexico City Metropolitan Area (MCMA) in the analysis presented below.

We also use Level 3 $NO_2$ tropospheric column data from the Aura Ozone Monitoring Instrument (OMI) to evaluate spatiotemporal patterns of $NO_2$ (Boersma et al., 2011) We have used the Quality Assurance for Essential Climate Variables (QA4ECV) $NO_2$ Level 3 product described in Boersma et al. (2018). This is the most recent product and has been improved over DOMINO v2. OMI $NO_2$ monthly mean data used in this study can be found via the TEMIS database (Boersma et al., 2017). The OMI $NO_2$ data is processed consistently with the QA4ECV formaldehyde (HCHO) product also used in this 230 study. We use the NASA Level 3 monthly mean tropospheric column HCHO measurements, also from the TEMIS database (De Smedt et al., 2017). Both L3 products are provided on a global 0.125° x 0.125° grid.

We also use observations of fires from the Moderate Resolution Imaging Spectroradiometer (MODIS) on both the Terra and Aqua satellites for fire counts and fire radiative power (FRP) to assess the seasonal peak in local fire activity. The MODIS 235 Active Fire product is provided by the Fire Information for Resource Management Systems (FIRMS) (doi:10.5067/FIRMS/MODIS/MCD14ML). The data is processed by MODIS Adaptive Processing System (MODAPS) using the enhanced contextual fire detection algorithm into the Collection 5 Active Fire product. Algorithm description for the MODIS Active Fire product can be found in Roy et al. (2008). There are two MODIS instruments in orbit, one on the NASA Terra satellite and one on the NASA Aqua satellite. Both Terra and Aqua fly in the NASA A-train constellation with 240 local overpass times of about 01:30 and 13:30. MODIS fire data has 1 km resolution. MODIS active fire products are not available for May 2021 on the FIRMS archive; thus we also use observations of fires from the S-NPP Visual Infrared Imaging Radiometer Suite (VIIRS) (doi:10.5067/FIRMS/VIIRS/VNP14IMGT_NRT.002). A comparative strength of VIIRS is the ability to detect smaller fires (Li et al., 2018; Wei et al., 2018), leading to a consistently higher fire count than MODIS. MODIS and VIIRS FRP data were filtered using only retrievals with >80 % confidence level.

## 3 Results and discussion

### 3.1 Spatial distribution of PANs over Mexico City


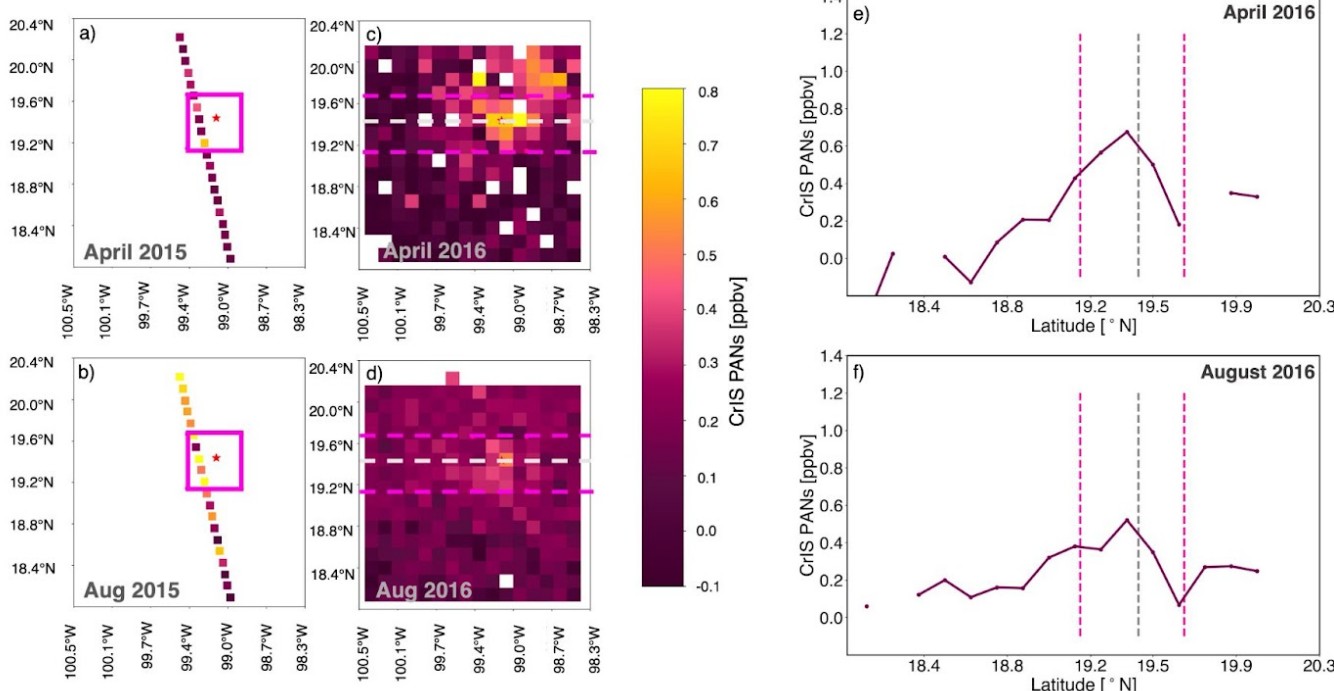

**Figure 3.** Maps demonstrating PANs enhancements around urban Mexico City. Monthly mean TES transects (n = 2) are shown for (a)
April and (b) August 2015, and monthly averaged gridded CrIS PANs are shown for (c) April and (d) August 2016. CrIS PANs are
plotted as a function of latitude for (e) April and (f) August. The latitude bounds of the urban box shown in panels a-b and Fig. 1 are
denoted by pink dashed lines in panels c-f. The Mexico City center is denoted by gray dashed lines.

Figure 3 presents the spatial extent of the PANs enhancement around urban Mexico City. April and August are plotted
because these are the two seasonal maxima in MCMA free tropospheric PAN from TES (not shown) and PANs from CrIS
(see Fig. 6 and later discussion). Figure 3a and 3b show the spatial extent of MCMA PAN from monthly averaged TES
transects (n = 2). The limited spatiotemporal and irregular temporal sampling by TES combined with the day-to-day
variability in PAN limits the utility of monthly averages within the TES dataset for individual months. Figure 3c and 3d
show clear PANs enhancements in monthly mean gridded CrIS data. Figure 3e and 3f show north-to-south slices of the data
presented in Fig. 3c and 3d. These panels show that the spatial extent of free tropospheric average PANs > 0.2 ppbv covers
0.5 degrees latitude (~50 km) in April. This is less well-defined in August, but this distance is comparable to the diameter of
the Mexico City basin (Molina et al., 2009). Thus Fig. 3 demonstrates that the spatial resolution of CrIS is sufficient to
determine the size of urban tropospheric PANs enhancements during specific seasons. Figure 3e and 3f also confirm that
there is a localized peak in free tropospheric PANs collocated with the center of Mexico City, consistent with local
production of PANs within the MCMA city limits.





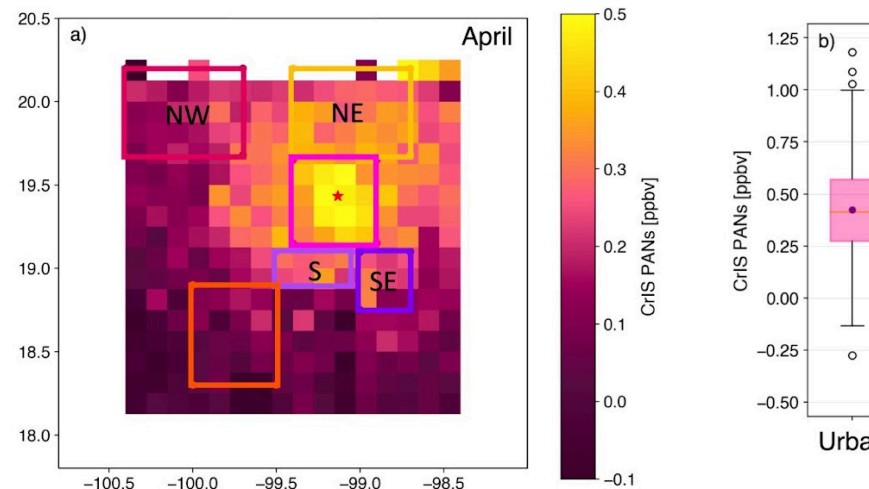
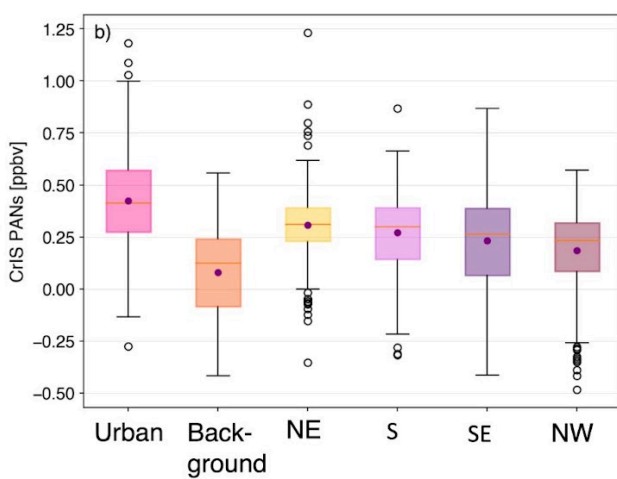

**Figure 4.** (a) Potential air pollutant outflow regions for the Mexico City basin (colored squares) plotted over monthly mean gridded CrIS PAN for April 2016-2021, **(**b) boxplots of daily CrIS free tropospheric PANs measurements within the boxes shown in (a). The orange lines and purple circles represent the median and mean respectively. The whiskers represent the standard deviation of daily CrIS free tropospheric PANs measurements.

The regional topography of the MCMA drives the meteorological transport patterns in the city. Previous studies show that thermally-driven mountain-valley flows contribute to most of the day-to-day variability in surface wind speed and direction, rather than circulations at the synoptic-scale (De Foy et al., 2005; De Foy et al., 2006, 2008; Doran & Zhong, 2000; Fast & Zhong, 1998; Lei et al., 2007, 2008; Zavala et al., 2020). Intense vertical shear of these thermally-driven flows lead to recirculation of air within the MCMA basin. Near-surface convergence zones due to momentum down-mixing of thermally-driven wind and light synoptic winds aloft can lead to the accumulation of pollutants in the basin. However, the MCMA basin has relatively effective venting through the Chalco passage to the southeast and northern plateau leading to little day-to-day pollutant accumulation (Zavala et al., 2020). Prior work has shown that on average the air quality "footprint" of MCMA is fairly local as air transported out of the basin is diluted quickly (e.g., Emmons et al., 2010; Mena-Carrasco et al., 2009). However, PANs are an exception because they can act as reservoirs and sources of $NO_x$ to increase downwind $O_3$ production. For example, during March 2006 a plume containing elevated $NO_y$ species was observed ~900 km downwind in the northeast outflow direction (Mena-Carrasco et al., 2009). Next we explore PANs in the most common outflow regions around Mexico City

Figure 4a denotes sub-regions within the CrIS data processing area around Mexico City. The regions denoted by NW, NE, S, and SE represent potential pollutant outflow regions for Mexico City, and our choice of regions is informed by the common patterns of pollutant outflow identified by De Foy et al. (2006) for the 2006 MILAGRO (Megacity Initiative: Local and Global Research Observations) field intensive. Figure 4b presents boxplots of daily CrIS free tropospheric PANs values within these regions. Together, these two panels show that there are more days with higher PANs in the region located to the northeast of Mexico City (yellow box) and in the region that encompasses the mountains south of the city (light purple box). These two regions appear to be the dominant directions of PANs outflow in MCMA in April. Our analysis implies that outflow of PANs to the northwest (magenta box) is infrequent. The largest spread in daily average PANs is over the southeast Chalco Passage (dark purple box). Figure 4 is largely consistent with the findings of De Foy et al. (2006; 2008) and Emmons et al. (2010) Pollutants can be transported out of Mexico City to the northeast, channeled through the southeast, or vertically injected out over mountains to the southwest, depending on meteorological conditions present.





## 3.2 Seasonal cycles of PANs over Mexico City

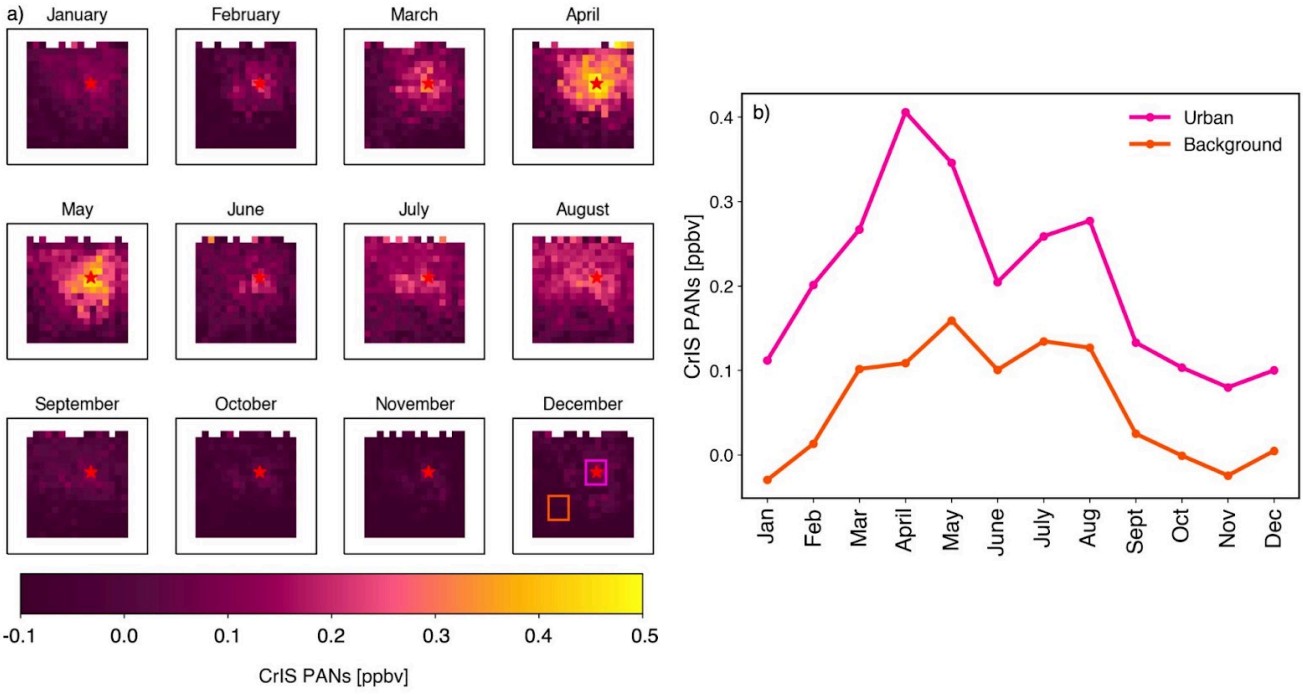

**Figure 5.** (a) Monthly mean gridded CrIS PANs over Mexico City. The boxes indicate the locations chosen to illustrate the urban Mexico City PANs (pink) enhancement compared to the "nearby background" (orange). The background box is confined by (18.99° N, -99.84° W, 18.49° N, -99.34° W) and the urban box is confined by (19.15° N, -99.40° W, 19.65° N, -98.9° W). (b) Monthly mean urban and background mean CrIS free tropospheric PANs for the period January 2016 to May 2021.

Figure 5 contrasts PANs over a region only rarely impacted by direct outflow from Mexico City to the PANs observed directly over Mexico City. Figure 5a presents monthly gridded mean PANs for the period 2016-2021. The orange box in the "December" panel surrounds an area with consistently low free tropospheric PANs mixing ratios. We refer to this area as "nearby background" PANs conditions here. Figure 5b presents a time series of monthly averaged CrIS PANs values within the two respective boxes shown in Fig. 5a. Based on this Figure, PANs are always greater over the urban area than in the "nearby background". On average, over the period 2016-2021, the difference between urban and background free tropospheric PANs is greatest in April and reaches a minimum in the winter months of October, November, and December. During these months, the average difference between urban and nearby background free tropospheric PANs is ~100 pptv. The nearby background PANs have a seasonal maximum in May, and this is consistent with the March-April-May peak in local fire activity (Yokelson et al., 2007; 2009; 2011).



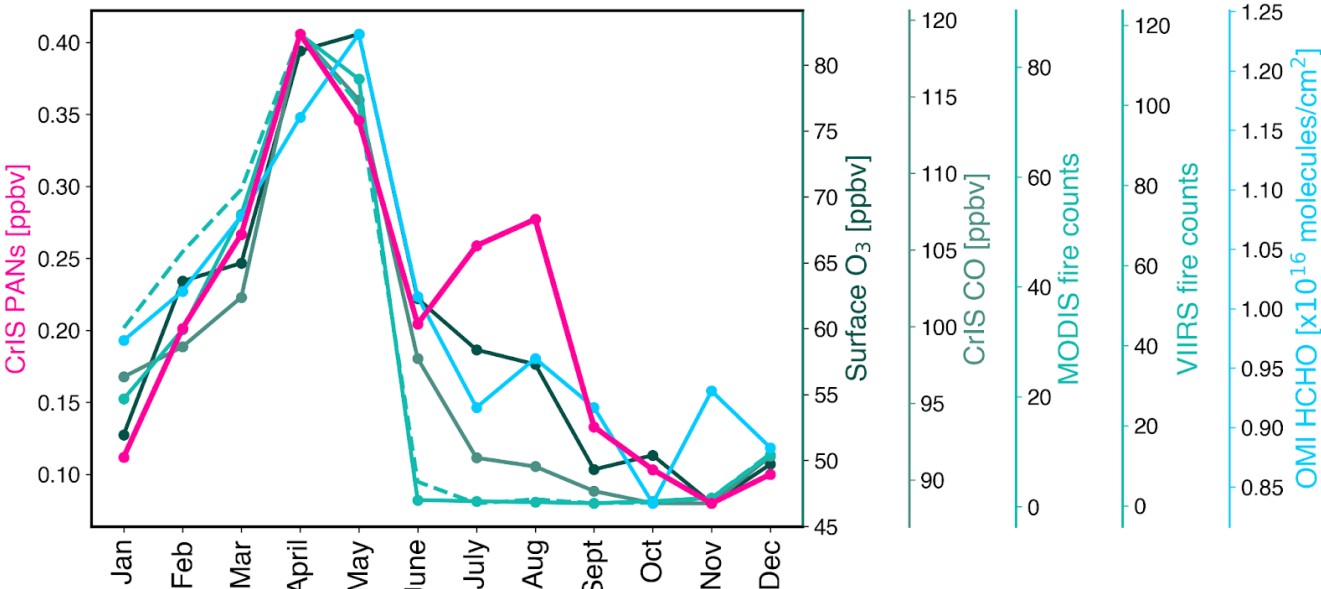

**Figure 6.** Seasonal cycle of CrIS free tropospheric PANs, surface $O_3$, CrIS free tropospheric CO, MODIS fire counts (solid; 2016-2020), VIIRS fire counts (dashed) and OMI HCHO columns for urban Mexico City from 2016-2021. MODIS and VIIRS fire counts are constrained by (20.12° N, -100.40° W, 18.36° N, -98.44° W). All other species are constrained by the urban box defined in Fig. 5.

Figure 6 displays the seasonal cycle of PANs from 2016-2021. The maximum in PANs over urban Mexico City occurs in April. Surface $O_3$, CrIS free tropospheric column mean CO, MODIS and VIIRS fire counts, and tropospheric column OMI HCHO have co-occurring peaks in March, April and May. Mexico City experiences a wet and a dry season. November through February is typically characterized by cool and dry conditions as anticyclonic westerly winds bring dry air to the region. This is followed by a warm, dry season from March to May, followed by a rainy season that typically lasts from May through October (Molina et al., 2009). The warm, dry season is characterized by high pressure systems with clear skies, weak wind, and intense solar radiation promoting photochemical production of $O_3$ and other oxidants (Molina et al., 2019). Weak winds during the warm, dry season promote stagnation of pollutants near the southern area of the basin (Molina et al., 2019; Zavala et al., 2020). The observed seasonal maxima in surface $O_3$ and tropospheric column OMI HCHO occur during this period (Fig. 6). Lei et al. (2009) show that secondary HCHO dominates the HCHO budget in the afternoon when the observations shown here were collected. Maxima in CrIS free tropospheric CO and MODIS fire counts occur in April and May. Local fire activity during these months contribute to increased local tropospheric CO abundances (Tzompa-Sosa et al., 2016; Yokelson et al., 2007; 2009; 2011). Note that the region included in Fig. 6 may not encompass all the fires that impact this location. It represents the local fire activity only. The rainy season is typically characterized by lower CO and $O_3$ mixing ratios; however, $O_3$ production continues throughout the year as intense photochemical reactions can occur prior to afternoon precipitation in the rainy season (Molina et al., 2019). Elevated $O_3$ mixing ratios can occur throughout the year in the MCMA due to its subtropical latitude and high elevation (Molina et al., 2009), though over 60 % of the $O_3$ episodes in Mexico City (i.e., exceedances to the 1-hour standard of 95 ppbv) occur during the warm, dry season (see Jaimes-Palomera et al., 2016 and references therein). Note that there is a secondary peak in PANs in July and August that is not associated with a similar increase in surface $O_3$. This suggests that the PAN observed by CrIS over this region may not be associated with local surface photochemistry. Model results included in the Supplemental Information (Figure S1 and S2) indicate that there are multiple $NO_x$ sources that contribute to this feature, and the PAN enhancement is located higher aloft (i.e. at 400 hPa versus 750 hPa).


## 3.3 Interannual variability in PANs over Mexico City

**Figure 7.** (a) Gridded average PANs for the month of May for the region surrounding Mexico City for multiple years. The center of Mexico City is denoted by a red star. (b) MODIS fire radiative power (FRP) for the Mexico City region, (c) VIIRS fire radiative power (FRP) for the Mexico City region, and (d) gridded free tropospheric average CrIS CO. This is the full region of available CrIS observations (21° N, -100.9° W, 18° N, -97.1° W).

In addition to air pollution from the anthropogenic sources, air quality in the MCMA is impacted by local (Yokelson et al., 2007) and regional (Yokelson et al., 2009) biomass burning, as well as agricultural residue and trash fires (Christian et al., 2010; Yokelson et al., 2011). March-April-May is the seasonal peak in regional biomass burning (Tzompa-Sosa et al., 2016; Yokelson et al., 2007; 2009; 2011). Biomass burning in Mexico and urban emissions from the MCMA heavily influence the springtime air quality in much of Mexico and the United States (Yokelson et al., 2009). Fires can contribute to a quarter of the CO production within the MCMA (Yokelson et al., 2007). Figure 7 displays monthly averaged gridded CrIS PANs (a) and CO (d) for May 2016 - 2021 and corresponding fire-related products in Fig. 7b and 7c. We focus on the month of May to



demonstrate the interannual variability in free tropospheric PANs over Mexico City during this month and their relationship to local fires. The free tropospheric PAN enhancement in May 2017 located south of the urban enhancement is collocated with fires. There were 196 and 234 fires detected in the region during May 2019 by MODIS and VIIRS, respectively. S-NPP CrIS is missing data between 26 March 2019 and 24 June 2019. A hardware failure occurred in the S-NPP CrIS mid-wave

signal processor, which led to a temporary halt in the L1b processing, but the issue was resolved by switching from the Side 1 electronics to the redundant Side 2 electronics. Note that JPSS-1 CrIS radiances are available over this time period, but targeted TROPESS/MUSES CrIS processing over megacities has not been performed for JPSS-1 at this time. The smallest number of fires were detected in May 2018 (N = 16; 33), and the corresponding average free tropospheric PANs and CO are both also lower than other years, though small enhancements in both exist collocated with fires to the south of the city. The

relationships between monthly mean CO and VIIRS ($R^2$ = 0.69) and MODIS fire counts ($R^2$ = 0.78) are strong. We also found that total monthly fire counts (plotted in Fig. 7b and 7c) explain a substantial amount (> 50 %) of the variability in monthly mean PAN during May; MODIS fire counts, using only 4 months of data, explain more of the variability in monthly mean PAN ($R^2$ = 0.75; Fig. 7b) than VIIRS fire counts, which include all 5 months of available data ($R^2$ = 0.52; Fig. 7c).



## 380   3.4 Recent changes in PANs, NOₓ, and HCHO

**Figure 8. (a)** Monthly average of OMI NO₂ (Dark Blue) and RAMA surface NOₓ observations (light blue), (b) CrIS free tropospheric monthly average PANs, (c) OMI HCHO tropospheric columns for the period 2016-2021. Distribution of OMI NO₂ (d), CrIS PANs (e), and OMI HCHO (f) from two populations: 2016-2017 (orange) and 2018-2021 (purple). Sample means (μ) are provided in their respective 385   colors and units.

In response to substantial air quality degradation the Mexican government developed and implemented successive air quality management programs that combined regulatory actions with technological advances (Molina, 2021; Molina et al., 2019). Reductions in O₃ were attributed to aggressive emission controls of O₃ precursor species, including improving fuels, 390   adopting catalytic converters, removing an oil refinery and heavy industrial facilities, shifting to natural gas for power generation and other industrial needs, reformulating liquefied petroleum gas (LPG) for cooling and water heating,





introducing vehicle inspection and maintenance programs, and introducing a "no driving day" (*Hoy no Circula*) rule (Zavala et al., 2020 and references therein). These control strategies led to reductions in all criteria pollutants (Zavala et al., 2020 and references therein); however, the MCMA basin still experiences elevated levels of many of the air pollutants listed above
(Cady-Pereira et al., 2017; Fast & Zhong, 1998; Gaffney et al., 1999; Jaimes-Palomera et al., 2016; Johansson et al., 2009; Molina et al., 2010; Osibanjo et al., 2021; Velasco et al., 2007; Zavala et al., 2020). There have been no substantial improvements in the concentration of $O_3$ (or $PM_{2.5}$ and $PM_{10}$) since ~2006 (Molina et al., 2019; Zavala et al., 2020). The transport sector continues to have a large impact on VOC and $NO_x$ emissions in the MCMA. Solvents, fuel evaporation and leaks, are large sources of VOCs, dominating over biogenic VOC emissions (Molina et al., 2010; Velasco et al., 2007). The
distribution of sources are highly inhomogeneous throughout the MCMA (Jaimes-Palomera et al., 2016; Zavala et al., 2020).

Figure 8 presents a time series and histograms of monthly averaged $NO_x$ species, CrIS PANs, and tropospheric column OMI HCHO for the CrIS measurement record period (2016-2021) for urban Mexico City. There is a statistically significant decrease in MCMA OMI $NO_2$ between 2018 and 2019. We separate the OMI $NO_2$ into two sample populations to represent
the higher-value population (2018-2021; orange) and the lower-value population (2016-2017; purple). Figure 8d shows that the more recent data (purple) peaks towards lower values and the histogram representing the older population (orange) peaks towards higher values. This is further expressed in the sample means; the 2016-2017 and 2018-2021 means are 311 $\times 10^{13}$ and 246 $\times 10^{13}$ molecules cm$^{-2}$ respectively (standard deviation: 63, 52 $\times 10^{13}$ molecules cm$^{-2}$, respectively). These are statistically different. However, there is no statistically significant difference in the mean free tropospheric PANs as
observed by CrIS between 2016-2017 and 2018-2021. The sample means are both 0.19 ppbv (standard deviation: 0.09, 0.10 ppbv, respectively). We find there is also no significant difference in OMI HCHO between the two time periods. HCHO can be emitted directly and it is an intermediate product in the oxidation of many VOCs (Lei et al., 2009). MCMA has elevated levels of HCHO from both primary emissions and secondary photochemical formation (Baez et al., 1995; Garcia et al., 2006; Lei et al., 2009; Velasco et al., 2007). Lei et al. (2009) showed that secondary HCHO dominates the MCMA HCHO budget
in the mid-morning and afternoon; coinciding with ascending satellite overpasses. HCHO is used as an indicator of VOC emissions (De Smedt et al., 2008; Shen et al., 2019) and changes in VOC emissions may not mirror changes in $NO_x$ (Gao et al., 2017; Shen et al., 2019; Simon et al., 2015).

The lack of a change in PANs is not surprising. PAN can be more sensitive to NMVOCs than to $NO_x$ emissions (Fischer et
al., 2014), and the PANs observed by CrIS are not all formed from local anthropogenic $NO_x$ emissions (see Fig. S1 and S2). Surface PANs can also respond to changes in the environmental conditions that support photochemistry. For example, during the COVID-19 lockdown period, emissions of both $NO_x$ and VOCs were reduced by 60 % and 30 %, respectively in the Beijing area, but Qiu et al. (2020) showed enhanced levels of surface PAN in this region. They showed that this enhanced PAN (2-3 times the concentrations of the pre lockdown period) was the result of enhanced photochemistry, anomalous wind
convergence leading to the accumulation of precursor species and accelerated VOC oxidation (i.e. further enhancing local photochemistry), and anomalously high temperatures to the area during the study period.

## 4 Conclusions

We use CrIS data from 2016-2021 to investigate the spatial and temporal variability of PANs in Mexico City. This is the first detailed analysis of satellite PANs observations over a megacity.

1. We use a period with densely spaced TES observations of megacities from 2013-2015 to investigate the cities with the largest mean detected PANs from this period. The mean TES free tropospheric PAN mixing ratios for observations over Mexico City was 0.35 ppbv. Mexico City showed the highest mean PAN values of all the 19
megacities sampled by TES during this period.

2. S-NPP CrIS and TES observations show that PANs are enhanced around urban Mexico City with a strong seasonality. The largest difference between the urban enhancement and nearby background occurs in April and reaches a minimum in winter months (October, November, December). A seasonal maximum in PANs occurs in



April and May. The seasonal peak in Mexico City PANs co-occurs with springtime seasonal maxima in surface $O_3$, CrIS CO, MODIS fire counts, and tropospheric OMI HCHO. Seasonal maximums in local photochemistry and fire activity both contribute to the seasonal maxima in PANs.

3.  We find that extreme fire years are associated with higher monthly mean PANs than low-fire years. However,
missing observations in 2019, which was a severe fire year, make it difficult to fully quantify the effect of fires on the observed interannual variability of PANs over Mexico City during the month of May. JPSS-1 radiances are available for this time period but the JPSS-1 CrIS PANs retrievals for dense sampling over MCMA are not available at the time of this submission.

4.  We use S-NPP CrIS data to probe the spatial outflow pattern of PANs produced within urban Mexico City during April. We show that outflow occurs to the northeast of the city and over the mountains south of the city. Outflow to the northwest appears infrequently.

5.  We examine time series of surface $NO_2$ measurements alongside OMI tropospheric $NO_2$ to analyze changes to
Mexico City $NO_2$ during our study period. We find a statistically significant difference (decrease) in $NO_2$ between 2018 and 2019. CrIS PANs for the same period do not show the same significant decrease, nor does tropospheric OMI HCHO. These results suggest that PAN is VOC-limited and not $NO_x$-limited in the Mexico City basin.

6.  We see a secondary peak in PANs in July and August that is not associated with an increase in surface $O_3$. Analysis
of model results indicate that there are multiple $NO_x$ sources that contribute to this feature, and the PAN enhancement is located higher aloft (i.e. at 400 hPa versus 750 hPa).

The work presented here provides new information on the seasonality and export pathways out of the Mexico City
metropolitan area (MCMA) for a globally-relevant pollutant. The analysis approach applied here has the potential to be applied to different megacities and regions of interest.

**Data Availability**

The full dataset for CrIS PANs and CO megacity data used in this study are available upon request to
Vivienne.h.payne@jpl.nasa.gov. A modified version used to make the figures in this paper can be downloaded at https://datadryad.org/stash/share/sWnPVIrj6k4FYqfOrIzybFKb-D-7RQ7MqipCH3sasCU or found at at 10.5061/dryad.547d7wmc9.
All other datasets are publicly available as noted:
TES PAN data were obtained from https://doi.org/10.5067/AURA/TES/TL2PANNS_L2.007 (NASA/LARC/SD/ASDC,
2017). RAMA data were obtained from http://www.aire.cdmx.gob.mx/ (RAMA, 2021). OMI $NO_2$ and HCHO data were obtained from https://www.temis.nl/airpollution/no2.php and https://h2co.aeronomie.be/ (Boersma, 2017; De Smedt, 2017). MODIS Collection 6 Hotspot / Active Fire Detections MCD14ML distributed from NASA FIRMS. Available on-line https://earthdata.nasa.gov/firms. doi:10.5067/FIRMS/MODIS/MCD14ML. NRT VIIRS 375 m Active Fire product VNP14IMGT distributed from NASA FIRMS. Available on-line https://earthdata.nasa.gov/firms.
doi:10.5067/FIRMS/VIIRS/VNP14IMGT_NRT.002. Data used to create the elevation map is from SRTM and is available at https://srtm.csi.cgiar.org (Jarvis, 2008).






**Author contribution**

EVF, VHP and SSK were responsible for the initial study design. MJS led the data analysis and writing with guidance from EVF, VHP and SSK. VHP and SSK were responsible for algorithm development. KM was responsible for model simulations. All authors edited the manuscript.

**Competing interests**

The authors declare that they have no conflict of interest.

**Acknowledgements**

This work is funded under NASA award number 80NSSC20K0947. Part of this work was carried out at the Jet Propulsion
Laboratory, California Institute of Technology, under a contract with the National Aeronautics and Space Administration (80NM0018D0004). We acknowledge the use of data and/or imagery from NASA's Fire Information for Resource Management System (FIRMS) (https://earthdata.nasa.gov/firms), part of NASA's Earth Observing System Data and Information System *(EOSDIS).* We thank Isabelle De Smedt and Folkert Boersma for collaborating on the proposal that led to this work and for answering questions regarding the OMI data.

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
