# Peer review of "Measurement Report: Spatiotemporal variability of peroxy acyl nitrates (PANs) over Mexico City from TES and CrIS satellite measurements"

_Atmospheric Chemistry and Physics, 2022_

## Referee Comment (RC1)

This manuscript utilized two satellite observations of PANs over Mexico City and analyzed the temporal and spatial variation of PANs. The measurements of other species (O3, CO, NO2 column, HCHO column) and fire activities are also included to interpret the seasonal trends of PANs. The paper is well-written, and the context is within the scope of a measurement report. I recommend minor revision before publication.

My major comment is the relationship between satellite observed PANs and local pollution in urban regions. The introduction section tells us how PANs concentration is elevated given the abundance of precursors. For cities, as the emissions are near the surface and the most pollution is within the mixing layer, we expect increased PANs within the mixing layer as well. However, both TES and Crsl PAN retrievals provide the free-tropospheric averages of PAN. Please justify in the context that these retrievals reflect PAN signals due to pollution in cities.

I also ask for clarification on few questions listed below:

1. Figure 3: As TES measures PAN only and CrIS measures all PANs, why TES PAN retrieval is higher than CrIS PANs retrievals in Figure 3 b) and d)?

2. Line 295 and Figure 4: The authors attribute the lowest PANs in NW region to infrequent outflow to the northwest. However, distances between the city center and selected outflow regions are different. The lowest PANs in NW can be the result of chemical/physical loss due to longer transport. Please clarify that.

3. Figure 5: The nearby background PANs are equally high between March and Aug. As the authors attribute the high PANs to local fire activity in March-April-May, the high PANs between June and August are not explained.

4: Line 349: Please specify that multiple NOx sources include anthropogenic, soil, biomass burning and lightning. It is also worth quantifying the relative contribution of each NOx source in the main context.

5. Figure 7: it is very hard to tell the difference of spatial patterns of CrlS PANs among years and how it correlates with the occurrence of fires. For instance, Line 367 says PAN enhancement in May 2017 located south of the urban enhancement is collocated with fires. However, the most fire occurs southwest of the domain, but no PANs enhancement is observed.

---

## Author Comment (AC1)

**Manuscript number ACP-2022-582**
**Response to Reviewer #1**

We thank both reviewers for their helpful feedback. We have addressed all comments. Our responses are below each suggestion in purple. Please note that we also applied a correction applied to CrIS PANs as suggested by Payne et al. [2022] to correct for a water vapor-dependent bias. The changes are very modest, but this brings this paper in line with Payne et al. [2022].

**Reviewer 1:**

This manuscript utilized two satellite observations of PANs over Mexico City and analyzed the temporal and spatial variation of PANs. The measurements of other species ($O_3$, CO, $NO_2$ column, HCHO column) and fire activities are also included to interpret the seasonal trends of PANs. The paper is well-written, and the context is within the scope of a measurement report. I recommend minor revision before publication.

My major comment is the relationship between satellite observed PANs and local pollution in urban regions. The introduction section tells us how PANs concentration is elevated given the abundance of precursors. For cities, as the emissions are near the surface and the most pollution is within the mixing layer, we expect increased PANs within the mixing layer as well. However, both TES and CrsI PAN retrievals provide the free-tropospheric averages of PAN. Please justify in the context that these retrievals reflect PAN signals due to pollution in cities.

We thank the reviewer for these helpful suggestions. We have addressed these comments below.

I also ask for clarification on few questions listed below:

**1. Figure 3: As TES measures PAN only and CrIS measures all PANs, why TES PAN retrieval is higher than CrIS PANs retrievals in Figure 3 b) and d)?**

Thank you for this comment. Clarification has been made within the Methods section of this work.

The TES and CrIS measurements shown here are from different time periods (2015 vs 2016). The CrIS retrievals used in this work were produced using NASA L1B full spectral resolution (FSR) radiances. The CrIS FSR L1B product is not available for the time period that overlaps with the TES megacity transect observations, and so direct comparison is not possible. There are other reasons why the TES PAN could be biased high relative to CrIS PANs. The TES and CrIS retrievals utilize different spectral regions (as described in the "Methods" section of this work). Based on retrievals using ground-based FTIR measurements covering both spectral regions, Mahieu et al. (2021) noted that PAN values retrieved using the 790 $cm^{-1}$ spectral feature used for the CrIS retrievals were biased low relative to those using the 1150 $cm^{-1}$ spectral feature used for the TES retrievals. As noted in the Methods section, the CrIS retrievals

are done in linear volume mixing ratio (as opposed to log vmr for TES retrievals). This could be an additional reason for differences.

Note that the CrIS PANs retrievals have been validated against aircraft data from the ATom campaigns (Payne et al., 2022). Fischer et al. (2018) provide a comparison between TES PAN transect observations coincident with Front Range Air Pollution and Photochemistry Éxperiment (FRAPPÉ) observations. This comparison showed that TES can have some degree of sensitivity to PAN in the boundary layer when boundary layer PAN is elevated, but no extensive validation of the TES retrievals has been possible, due to the relatively sparse spatial coverage of TES.

**2. Line 295 and Figure 4: The authors attribute the lowest PANs in NW region to infrequent outflow to the northwest. However, distances between the city center and selected outflow regions are different. The lowest PANs in NW can be the result of chemical/physical loss due to longer transport. Please clarify that.**

Thank you, this has been clarified in the manuscript. The box denoting outflow to the NW was chosen to represent an area not influenced by the surrounding mountains (see Figure 1). The lower values of PANs in the NW region could also be attributable to loss due to longer transport distance.

**3. Figure 5: The nearby background PANs are equally high between March and Aug. As the authors attribute the high PANs to local fire activity in March-April-May, the high PANs between June and August are not explained.**

Figure 5b shows a time series of average PANs in the box we defined to be not influenced by urban production ("background") in orange. This shows a maximum in the orange line (background) in May, which is routinely influenced by local fire activity.

Enhancements to background PANs were not explored outside of May. The model results presented in the SI show elevated PANs aloft in JJA (Fig. S1).

**4: Line 349: Please specify that multiple NOx sources include anthropogenic, soil, biomass burning and lightning. It is also worth quantifying the relative contribution of each NOx source in the main context.**

As suggested by the reviewer, we have moved this information from the SI to the main text.

**5. Figure 7: it is very hard to tell the difference of spatial patterns of CrIS PANs among years and how it correlates with the occurrence of fires. For instance, Line 367 says PAN enhancement in May 2017 located south of the urban enhancement is collocated with fires. However, the most fire occurs southwest of the domain, but no PANs enhancement is observed.**

The below figure includes correlation plots of monthly mean PANs, CO, and fire counts whose $R^2$ values have been reported in this section.

[Figure]

**Manuscript number ACP-2022-582**
**Response to Reviewer #2**

**Reviewer 2 comments:**

Madison et al., 2022., ACPD, Measurement Report: Spatiotemporal variability of peroxy acyl nitrates (PANs) over Mexico City from TES and CrIS satellite measurements.

**General description of the manuscript:**

In this measurement report, the authors demonstrate the spatiotemporal variability of PANs over Mexico City and its surrounding area using the space-based observations from the Tropospheric Emission Spectrometer (TES) and the Suomi National Polar-orbiting Partnership (S-NPP) Cross-Track Infrared Sounder (CrIS) instruments and exploring the spatial outflow pattern of PANs produced within urban Mexico City during the seasonal maxima.

**General Comments:**

**Overall, this is an interesting work. I suggest it be considered for publishing subject to some of the technical comments mentioned below.**

Thank you for your comments. We have implemented the technical comments into the revised manuscript.

**Abstract:**

**State the observation years of the study.**

**Line 19 : Please define the Mexico City spring.**

This has been fixed. Thank you.

**Line 22: The authors should expand the abbreviation(NW) for the first time.**

This has been fixed. Thank you.

**Line 80: Include references for TES and CrIS.**

Both datasets are referenced in the section detailing data description.

**Line 204: Can you add a reference to justify the screening? Is it applicable for both CO and PAN retrievals? How does PAN retrieval differ from Payne et al. (2022)**

This PAN retrieval is the same presented in Payne et al. (2022) as stated in the methods section of the text; "Details of this retrieval can be found in Payne et al. (2022)...".

Clarification has been added to the text: "To be included in this analysis, CrIS retrievals have to be deemed "good" by the respective master quality flags (CO and PANs), and retrievals have to meet the additional following criteria as recommended by the developers of these data (Payne et al., 2022). The first four flags are standard quality flags, and the fifth checks whether the PANs spectral shape was fit:"

**Line 240: Correct the overpass time of Terra and Aqua satellites.**

Corrected. Thank you.

**Line 250; Figure 3: What do the white pixels on the maps represent? You can change the color scale to include the NaNs or missing data. Monthly mean TES transects (n = 2)… what does the 'n' indicate?**

This has been clarified in the figure caption. This now reads: "White pixels show grid boxes with data that has been filtered out". TES transects n = 2 indicates 2 transects within the monthly average. This is clarified in the main text following Figure 3.

**Line 255 & 253: The author uses 'urban Mexico City' and 'urban box'. Please be consistent. Is the 'center of Mexico City' same as MCMA (as in line 38)? Please define them clearly and be consistent.**

Thank you. This has been fixed for consistency in the main text. This area will always be referred to as "urban Mexico City". MCMA in line 38 is defining the acronym, referring to a different, more general area than the "urban Mexico City" used for our analysis.

**Line 263: Please ensure consistency Fig/Figure.**

The journal requires a different format depending on where Fig/Figure lands in the sentence.

From ACP guidelines for submission:

- "The abbreviation "Fig." should be used when it appears in running text and should be followed by a number unless it comes at the beginning of a sentence, e.g.: "The results are depicted in Fig. 5. Figure 9 reveals that...".".

**Figure 4: Missing labels on the map: for the X-Y axes and the 'Background'& 'Urban' boxes.**

These boxes have been defined in previous figures (Fig. 1 and Fig. 3) and are defined in 4b.

**Figure 5: Use either '-ve' or 'W' for the longitude.**

Thank you. This has been corrected.

**Is "nearby background" in Figure 5 the same as "background' region in Figure 4?**

Correct. This has been clarified in the text as per this recommendation.

**Figure 6: You may include the surface O3 data source (RAMA?) to match with other data in the caption.**

Thank you. This has been added.

**Line 317: During these months.. More specific here.**

Thank you. This has been clarified. This now reads: "During these **winter** months"

**Line 325: Is there any specific reason for choosing a different/larger region for the fire counts?**

Wildfires will typically not occur directly within urban areas but smoke can impact urban areas. The larger box was includes nearby regions with fires detectable from satellite.

**Line 390-340: Not very clear. Please reword the sentence.**

Wording of sentences has been checked for clarity. This now reads (now lines 437-440): "Reductions in $O_3$ were attributed to aggressive emission controls of $O_3$ precursor species. This included improving fuels, adopting catalytic converters, removing an oil refinery and heavy industrial facilities, shifting to natural gas for power generation and other…"

**Line 339 – 340: Note that the region included in Fig. 6 may not encompass all the fires that impact this location. Which location are you mentioning here? Is it the urban box in Figure 5 and Figure 6?**

The location being referred to here is the larger region surrounding Mexico City. This has been clarified in the text. Thank you.

**Figure 8: Missing x-labels for the figure. Histograms (panels (d-f) )are not consistent in bin size. Please define counts.**

The X label for the figure is at the top of the figure. Histograms do not have consistent bin sizing because the datasets have different distributions. The counts refer to counts of data within that y value.

**Line 421: …the PANs observed by CrIS are not all formed from local anthropogenic NOx emissions. It would be nice to add the percentage contributions from the NOx sources.**

Unfortunately we do not have the data to do this. Model simulations only exist for 2017. Thus model and CrIS observations are not directly comparable over the entire time period.

**Line 454: Replace surface NO2 by surface NOx**

Great catch. Thank you!

**Authors find that the Seasonal maximums in local photochemistry and fire activity contribute to the seasonal maxima in PANs (in conclusion, line 441). However, there is a contradictory statement that the PAN observed by CrIS over this region may not be associated with local surface photochemistry (line 347). The authors should clarify this contradiction.**

Line 347 is referring to PANs aloft during the August 2nd maxima. Line 441 is referring to April maxima at the surface. This has been clarified in what is now line 383.

Line 383-384: This suggests that the PANs observed by CrIS over this region may not only be associated with local surface photochemistry but PANs aloft (Fig S1).

**Do the authors check the directions of PANs outflow in MCMA for other seasonal maxima (e.g., August)? That would be strong evidence to draw a generalized conclusion of the directions of PANs outflow for seasonal maximums.**

Regions of outflow were chosen based on the MILAGRO 2006 campaign and De Foy et al., 2006; 2007; 2008 based in March, as discussed in section 3.1 of the text. Results of our analysis were similar for April and March. We choose to show April because the signal is enhanced by the maximum urban abundance of PANs, as shown by our seasonal cycles (Fig. 5 and Fig. 6).

August would not be an optimal month to analyze outflow because the magnitude of the urban enhancement is substantially less than that in April. Months with discrepancies between urban and background means (Fig. 5) would be other good candidates for this outflow analysis, but August would not.

**References**:

Fischer, E. V., Zhu, L., Payne, V. H., Worden, J. R., Jiang, Z., Kulawik, S. S., Brey, S., Hecobian, A., Gombos, D., Cady-Pereira, K., and Flocke, F.: Using TES retrievals to investigate PAN in North American biomass burning plumes, Atmos. Chem. Phys., 18, 5639–5653, https://doi.org/10.5194/acp-18-5639-2018, 2018.

Mahieu, E, et al. 2021. First retrievals of peroxyacetyl nitrate (PAN) from ground- based FTIR solar spectra recorded at remote sites, comparison with model and satellite data. Elem Sci Anth, 9: 1. DOI: https://doi.org/10.1525/elementa.2021.00027

Payne, V. H., Kulawik, S. S., Fischer, E. V., Brewer, J. F., Huey, L. G., Miyazaki, K., Worden, J. R., Bowman, K. W., Hintsa, E. J., Moore, F., Elkins, J. W., and Juncosa Calahorrano, J.: Satellite measurements of peroxyacetyl nitrate from the Cross-Track Infrared Sounder: comparison with ATom aircraft measurements, Atmos. Meas. Tech., 15, 3497–3511, https://doi.org/10.5194/amt-15-3497-2022, 2022.